# Discovery and structural characterization of monkeypox virus methyltransferase VP39 inhibitors reveal similarities to SARS-CoV-2 nsp14 methyltransferase

Jan Silhan [1,3], Martin Klima [1,3], Tomas Otava[1,2,3], Petr Skvara [1,3], Dominika Chalupska [1], Karel Chalupsky[1], Jan Kozic[1], Radim Nencka [1 ✉] & Evzen Boura [1 ✉]

Monkeypox is a disease with pandemic potential. It is caused by the monkeypox virus (MPXV), a double-stranded DNA virus from the *Poxviridae* family, that replicates in the cytoplasm and must encode for its own RNA processing machinery including the capping machinery. Here, we present crystal structures of its 2′-O-RNA methyltransferase (MTase) VP39 in complex with the pan-MTase inhibitor sinefungin and a series of inhibitors that were discovered based on it. A comparison of this 2′-O-RNA MTase with enzymes from unrelated single-stranded RNA viruses (SARS-CoV-2 and Zika) reveals a conserved sinefungin binding mode, implicating that a single inhibitor could be used against unrelated viral families. Indeed, several of our inhibitors such as TO507 also inhibit the coronaviral nsp14 MTase.

Monkeypox is a disease caused by the monkeypox virus (MPXV), an Orthopox virus belonging to the *Poxviridae* family, clinically resembling the eradicated smallpox[1]. Until recently, its occurrence was limited to central and western Africa where its natural reservoir consists of rodents and primates;[2] however, it can also infect humans, and the mortality rate is estimated to be 3–6%. This rate is lower than deadly smallpox, but much higher than COVID-19 caused by SARS-CoV-2. Monkeypox has become increasingly spread across the globe with the number of cases growing exponentially reaching a somewhat alarming 54,900 cases, and it has been reported in some 100 countries[3]. It is no wonder that the prospective of another viral pandemic has alarmed public health authorities as well as the general public.

MPXV is a dsDNA virus; however, it replicates in cytoplasm, implying that it must encode for DNA and RNA replication machinery[4] because the human enzymes are located in the nucleus. Besides the DNA-dependent DNA polymerase and the DNA-dependent RNA polymerase, it also encodes for the RNA capping machinery. In an early

stage of cellular infection, the RNA cap is important for initiation of the translation of the viral RNA (vRNA)[5,6]. This cap plays an important role in innate immunity, as uncapped RNA is recognized by the innate immunity IFIT and RIG-I sensors[7,8]. The cap is also important for stability of the mRNA, thus the poxviruses encode for the decapping enzymes to prevent accumulation of dsRNA later during infection which would induce the innate antiviral response[9]. Indeed, protection against innate immunity is of the uttermost importance to the MPXV, and it also encodes poxin, an enzyme that blocks the cGAS-STING pathway triggered by the presence of dsDNA in cytoplasm[10].

Chemically, the mRNA cap is a structure where an N7-methylated guanosine is linked via a triphosphate to the 5′ end of the RNA yielding m7GpppRNA, also referred to as cap-0. In the fully matured cap (cap-1), the first nucleotide is also methylated at the 2′-O position of the ribose moiety. In the *Poxviridae* family, cap-0 is synthesized in cytoplasm by the heterodimeric capping enzyme that has RNA 5′ triphosphatase (RTPase), guanylyltransferase (GTase) and

---

[1]Institute of Organic Chemistry and Biochemistry AS CR, Prague 6, Czech Republic. [2]Faculty of Food and Biochemical Technology, University of Chemistry and Technology, Prague 6, Czech Republic. [3]These authors contributed equally: Jan Silhan, Martin Klima, Tomas Otava, Petr Skvara. ✉e-mail: nencka@uochb.cas.cz; boura@uochb.cas.cz

(guanine-N7)-methyltransferase (MTase) activity[11]. The addition of another methyl group to the 2′-O position of the adjacent ribose converts cap-0 to cap-1. This step is also important in preventing an innate-immune response[12] and is catalyzed by the MPXV 2′-O MTase VP39[13].

Here, we present the structure of the MPXV VP39 in complex with the pan-MTase inhibitor sinefungin. The structure reveals the mechanism of VP39 inhibition and comparison of this structure to the 2′-O-MTases from unrelated viruses SARS-CoV-2 and Zika has important implications in the design of pan-antivirals based on MTase inhibitors.

## Results

### Overall structure of the MPXV VP39 MTase

To obtain the structure, we expressed and purified recombinant MPXV VP39 in *E. coli*. The recombinant protein was supplemented with sinefungin and screening of the crystallization conditions was performed. Initial crystals were of poor quality but diffraction quality crystals were obtained upon a few rounds of seeding. The crystals diffracted to 2 Å resolution and belonged to the orthorhombic $P2_12_12_1$ space group. The structure was solved by molecular replacement and refined to $R_{work} = 20.81\%$ and $R_{free} = 23.24\%$ and good geometry (detailed in Table 1). The structure revealed a mixed α/β fold that resembles the Rossmann fold. A central β-sheet is composed of β-sheets β2-β10 that adopt the shape of the letter *J*. Interestingly, this arrangement was also observed for the SARS-CoV-2 2′-O-MTase nsp16[14]. The central sheet is stabilized from one side by helices α1, α2, α6 and α7, and

by α3 and α7 from the other. These opposite sides of the *J*-like β-sheet are connected by α 5, β1 and β11 (Fig. 1).

We tested two substrates that differed in the penultimate base (m7GpppA-RNA vs. m7GpppG-RNA) to verify the enzymatic activity of our recombinant VP39 enzyme. Both substrates were accepted; however, the guanine base was clearly preferred in the penultimate position (Supplementary Fig. 1). That is in contrast to coronaviruses where adenine is preferred at this position[15].

### Analysis of the VP39-sinefungin interaction

The electron density for sinefungin was immediately visible upon molecular replacement in the central pocket of the VP39 MTase (Fig. 1). Sinefungin occupied the SAM (S-Adenosyl methionine) binding pocket (Fig. 2a). Its adenine moiety lies within a deep canyon lined with hydrophobic sidechains from residues Phe115, Val116, Val139 and Leu159 and makes two hydrogen bonds with the backbone of Val116 (Fig. 2b). The ribose and amino acid moieties are also involved in hydrogen bonding: the ribose ring directly binds Asp95 and Arg97 via its hydroxyl groups while the amino acid moiety forms hydrogen bonds with the sidechains of Gln39 and Asp138 (Fig. 2b) similarly as SAH (S-Adenosyl-L-homocysteine) (Fig. 2d).

Based on the previously solved crystal structure of vaccinia VP39 in complex with RNA[16], we constructed a model of a ternary complex sinefungin/RNA/VP39 to illustrate the molecular mechanism of sinefungin's action (Fig. 2c). It efficiently protects the 2′-O-ribose position; its amino group is in the close vicinity of the 2′ ribose hydroxyl group where the sulphur atom of SAM would be located otherwise.

## Table 1 | Data collection and refinement statistics

| Crystal | VP39-SFG | VP39-SAH | VP39-TO427 | VP39-TO494 | VP39-TO500 | VP39-TO507 | VP39-TO1119 |
|---|---|---|---|---|---|---|---|
| PDB code | 8B07 | 8CGB | 8CEQ | 8CER | 8CES | 8CET | 8CEV |
| Data collection | | | | | | | |
| Space group | $P\,2_1\,2_1\,2_1$ | $P\,2_1\,2_1\,2_1$ | $P\,2_1\,2_1\,2_1$ | $P\,2_1\,2_1\,2_1$ | $P\,2_1\,2_1\,2_1$ | $P\,2_1\,2_1\,2_1$ | $P\,2_1\,2_1\,2_1$ |
| Cell dimensions | | | | | | | |
| a, b, c (Å) | 51.9, 84.4, 153.8 | 52.3 84.1 154.9 | 51.7 83.9 154.4 | 51.9 84.0 154.4 | 51.6 84.1 154.1 | 52.0 84.1 154.3 | 52.3 84.5 155.1 |
| α, β, γ (°) | 90, 90, 90 | 90, 90, 90 | 90, 90, 90 | 90, 90, 90 | 90, 90, 90 | 90, 90, 90 | 90, 90, 90 |
| Resolution(Å) | 49.17–2.05 | 49.57–2.47 | 44.03–2.5 | 43.83–2.6 | 43.98–2.5 | 42.07–2.5 | 44.09–2.14 |
| | (2.13–2.05) | (2.56–2.47) | (2.59–2.5) | (2.69–2.6) | (2.59–2.5) | (2.59–2.5) | (2.22–2.14) |
| $R_{merge}$ | 0.273 (2.28) | 0.5451 (3.1) | 0.340 (2.98) | 0.108 (0.996) | 0.189 (1.24) | 0.150 (1.12) | 0.298 (2.54) |
| I / σ(I) | 9.82 (1.10) | 5.67 (0.82) | 6.02 (0.75) | 11.41 (1.34) | 8.47 (1.27) | 10.59 (1.36) | 7.97 (1.09) |
| $CC_{1/2}$ (%) | 99.6 (44.8) | 97.8 (42.3) | 98.2 (30.5) | 99.6 (72.6) | 99.0 (55.8) | 99.3 (71.0) | 99.5 (49.4) |
| Completeness (%) | 99.8 (98.5) | 99.9 (99.8) | 97.3 (94.0) | 86.0 (73.5) | 91.1 (78.5) | 87.2 (76.7) | 99.8 (99.6) |
| Redundancy | 13.1 (13.4) | 13.2 (13.7) | 7.1 (6.7) | 5.2 (4.1) | 6.2 (5.5) | 5.9 (4.8) | 13.2 (12.8) |
| Refinement | | | | | | | |
| No. reflections | 43138 (4205) | 25226 (2453) | 23387 (2241) | 18387 (1538) | 21839 (1864) | 21075 (1832) | 38698 (3788) |
| Rwork | 0.208 (0.315) | 0.201 (0.304) | 0.232 (0.352) | 0.216 (0.322) | 0.236 (0.306) | 0.219 (0.304) | 0.213 (0.302) |
| Rfree | 0.232 (0.331) | 0.252 (0.367) | 0.261 (0.384) | 0.251 (0.348) | 0.282 (0.402) | 0.272 (0.378) | 0.244 (0.348) |
| No. atoms | 5095 | 4868 | 4632 | 4673 | 4664 | 4670 | 4908 |
| Protein | 4686 | 4639 | 4527 | 4564 | 4548 | 4549 | 4601 |
| Ligand/ion | 54 | 46 | 68 | 76 | 74 | 76 | 54 |
| Water | 355 | 183 | 37 | 33 | 42 | 45 | 253 |
| B-factors | 37.11 | 40.30 | 48.72 | 46.73 | 41.05 | 40.29 | 40.62 |
| Protein | 37.05 | 40.55 | 48.85 | 46.83 | 41.18 | 40.40 | 40.83 |
| Ligand/ion | 35.25 | 48.33 | 35.53 | 34.88 | 30.14 | 31.98 | 38.32 |
| Water | 38.19 | 36.27 | 47.55 | 45.59 | 39.52 | 38.63 | 33.94 |
| R.m.s. deviations | | | | | | | |
| Bond lengths (Å) | 0.003 | 0.011 | 0.004 | 0.025 | 0.005 | 0.004 | 0.002 |
| Bond angles (°) | 0.53 | 0.80 | 0.60 | 0.52 | 0.67 | 0.55 | 0.47 |

Values in parentheses are for highest-resolution shell.

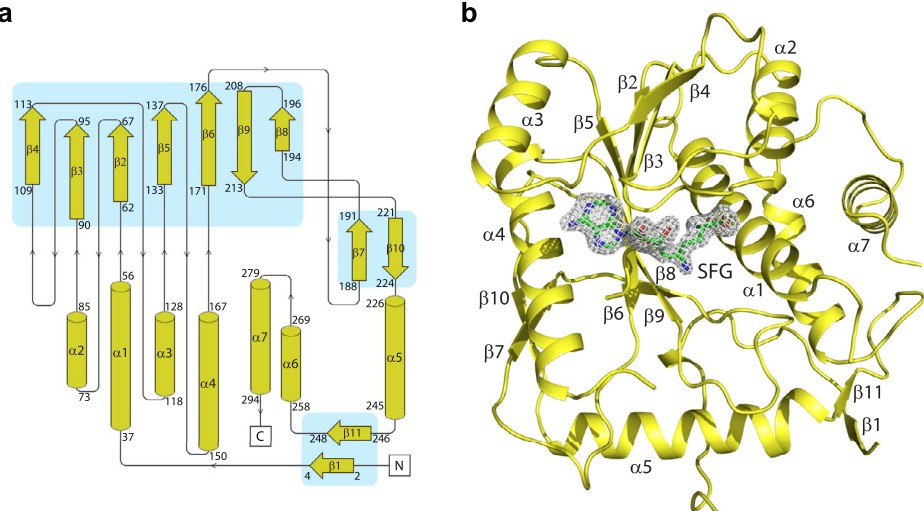

**Fig. 1 | Crystal structure of the monkeypox virus VP39 methyltransferase in complex with sinefungin. a** Topology plot of the monkeypox virus VP39 protein. **b** Overall fold of the VP39 protein in complex with sinefungin. The protein backbone is shown in cartoon representation and depicted in yellow, while sinefungin is shown in the stick representation and colored according to its elements: carbon, green; nitrogen, blue; oxygen, red. The Fo-Fc omit map contoured at 3σ is shown around sinefungin.

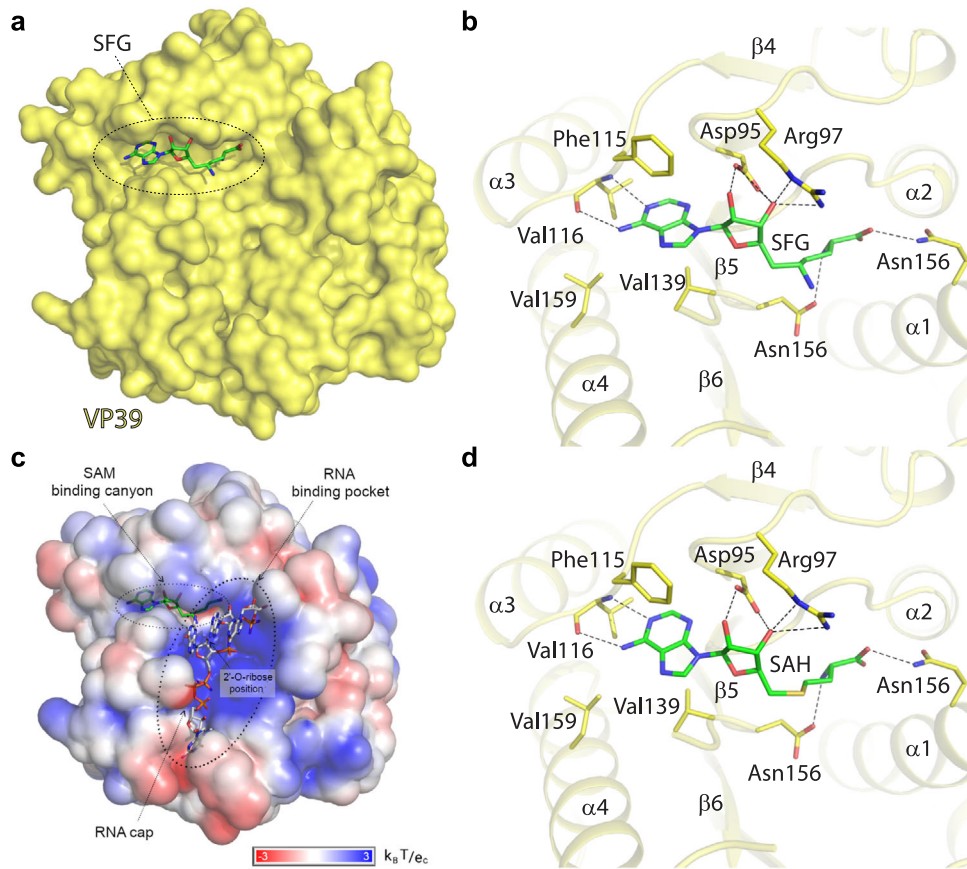

**Fig. 2 | Sinefungin and SAH recognition by the monkeypox virus VP39 methyltransferase. a** Sinefungin is bound in the SAM binding site. VP39 is shown in surface representation. **b** Detailed view of sinefungin or **d** SAH in the binding site. Sinefungin, SAH and side chains of selected VP39 amino acid residues are shown in the stick representation with carbon atoms colored according to the protein/ligand assignment and other elements colored as usual. Selected hydrogen bonds involved in the VP39-sinefungin interaction are presented as dashed black lines. **c** Model of RNA recognition by the monkeypox virus VP39 MTase, surface of the VP39 protein is colored according to the electrostatic surface potential. The m7GpppG-capped RNA was modeled by structural alignment using the crystal structure of the vaccinia virus VP39 protein in complex with SAH and m7GpppG-capped RNA (pdb entry 1AV6) as a template[16]. Sinefungin and m7GpppG-capped RNA are shown in the stick representation and colored according to the elements.

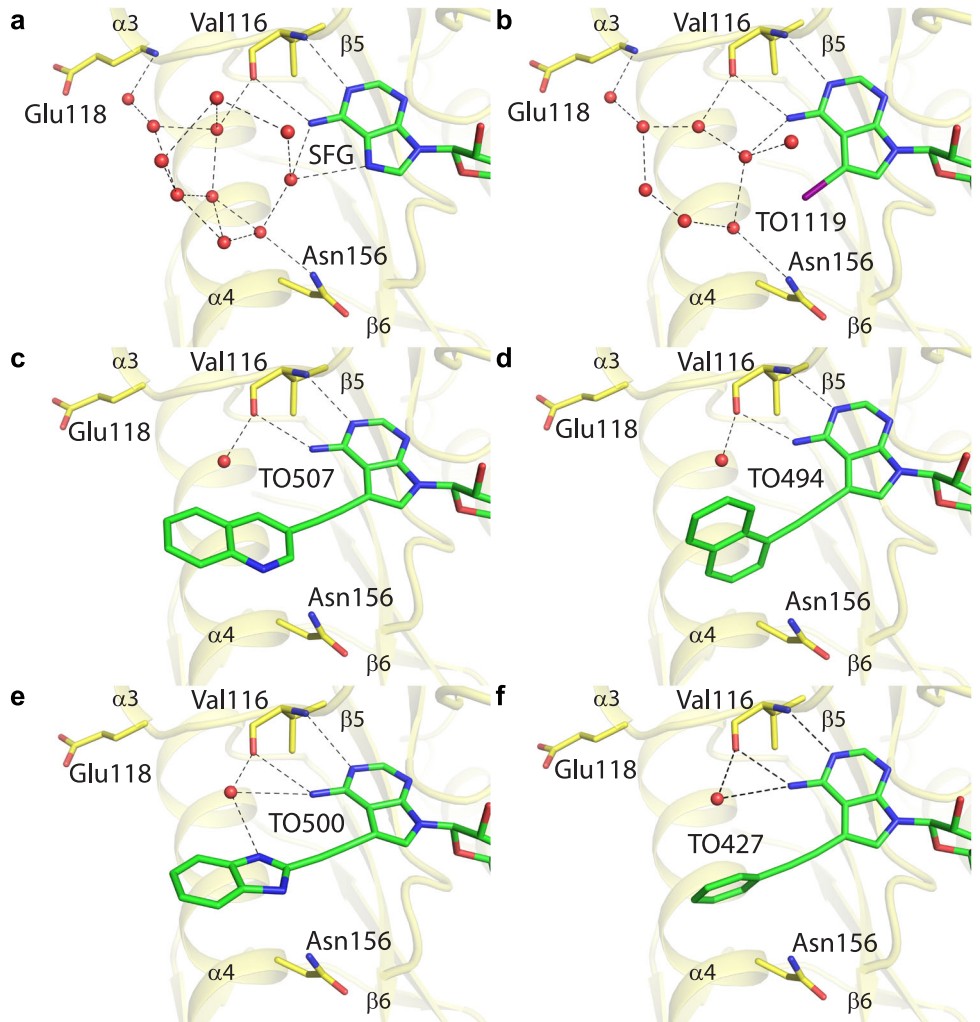

**Fig. 3 | A network of coordinated water molecules in the vicinity of the sine-fungin binding groove is disturbed by inhibitor binding. a** Detailed view of the water molecules coordinated next to the adenine base of sinefungin. Sinefungin and the side chains of selected VP39 amino acid residues are shown in the stick representation with carbon atoms colored according to the protein/ligand assignment and other elements colored as in Fig. 1b. Water molecules are shown as red spheres; selected hydrogen bonds are presented as dashed black lines. **b–f** The same view for different inhibitors.

## Inhibitor design

The SAM binding canyon has two ends: one borders the RNA binding pocket and serves to position SAM for the methyltransferase reaction; the other end adjacent to the adenine base of sinefungin is apparently unoccupied (Fig. 2). However, a close inspection of this location reveals a network of water molecules that are interconnected by hydrogen bonds and also bond the adenine moiety and residues Val116, Glu118 and Asn156 (Fig. 3a). Compounds based on the sinefungin scaffold bearing a moiety that would displace these waters and interact with Val116, Glu118 and Asn156 directly could be exceptionally good binders because displacement of water molecules can have a favorable entropic effect[17,18]. The best position to attach such a moiety is the N7 position of the adenine base. However, a 7-deaza purine base must be used to allow the introduction of suitable substituents. We screened a small in-house library of SAH analogs that contains compounds bearing large aromatic or heterocyclic substituents at this position. We also synthesized several new compounds where a small aliphatic chain is connected to the 7-deaza position.

We identified several compounds that inhibited the VP39 MTase significantly better than sinefungin (Fig. 4). Among them, three (TO1111, TO1116, TO1119) had a relatively small substituent (small aliphatic chain or iodine atom) and five (TO427, TO494, TO500, TO504,

TO507) had a large aromatic or heterocyclic substituent (Figs. 3 and 4). Surprisingly, the efficacy of these compounds was very similar (IC$_{50}$ = 0.08–0.17 µM) and an order of magnitude better than sinefungin (IC$_{50}$ = 2.3 µM).

## Binding mode of the inhibitors

To confirm our hypothesis that the inhibitors use the discovered water-filled cavity (Fig. 3a) we performed a crystallographic analysis. We used the same approach as with sinefungin and we were able to obtain well diffracting crystals for five compounds (TO427, TO494, TO500, TO507, TO1119). In each case the electron density for the inhibitor was well visible allowing us to unambiguously model the inhibitor although the electron density for the large 7-deaza substituents was less defined than that for the SAH scaffold of the molecule (Supplementary Fig. 2). This could be caused by free rotation of the substituent along the triple bond of the linker that would blur the density or by a small amount of SAH (~10–20%) left in the SAM binding pocket (recombinant MTases carry SAH from bacteria).

The crystallographic analysis revealed that in each case the moiety attached to the 7-deaza position is located within the water cavity. In the case of TO1119 that bears the large iodine atom at the 7-deaza position of the nucleobase only the water molecules near the iodine bound to the 7-deaza position were rearranged (Fig. 3b). However, the

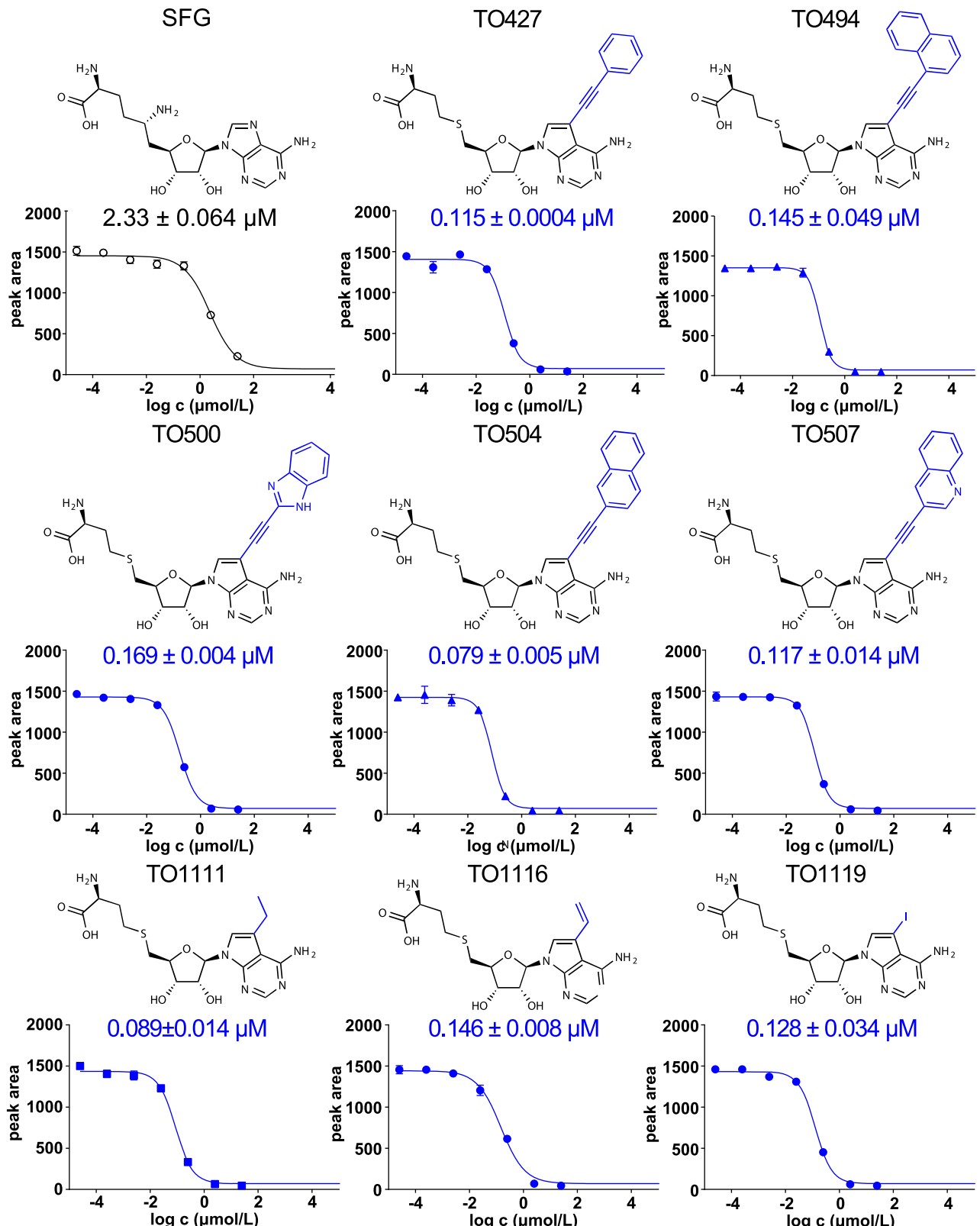

**Fig. 4 | Small molecule inhibitors of VP39 methyltransferase and their IC$_{50}$ values.** Structures of sinefungin (SFG) and small molecule inhibitors modified at the 7-deaza position used in this study (shown as blue lines), followed by IC$_{50}$ values and inhibition curves fitted to data from the MTase inhibition assay. Data points and IC$_{50}$ values are presented as mean values ± SEM ($n$ = 2 independent measurements). Source data are provided as a Source Data file.

inhibitors bearing bulky substituents at this position caused rearrangement of the majority of the water molecules, suggesting that the bulky moieties disrupted the water network within this cavity (Fig. 3c–f). TO1119 that only slightly disrupts the water network has

IC$_{50}$ = 128 nM while all the inhibitors with bulky substituents had a lower IC$_{50}$ except for the only substituent that is able to form a hydrogen bond, the TO500 that possesses a benzimidazole heterocycle. These results suggest that for optimal inhibitor binding it is

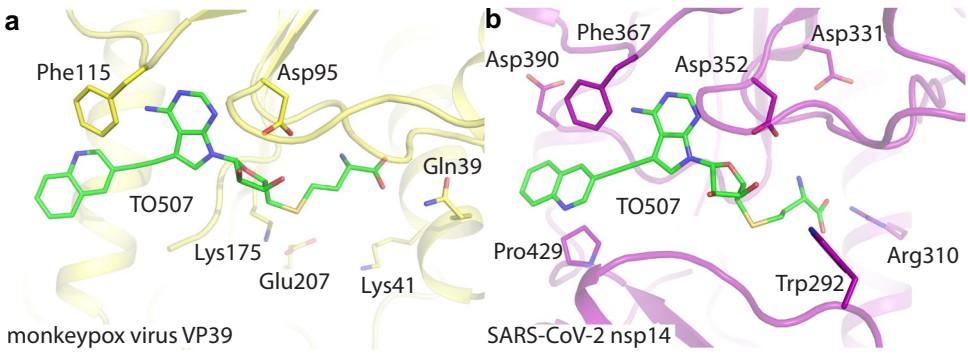

**Fig. 5 | Structural alignment of the MTase active sites of monkeypox virus VP39 with SARS-CoV-2 nsp14. a** Inhibitor TO507 in the SAM pocket of VP39. MTase. **b** Docking model of inhibitor TO507 bound to SARS-CoV-2 nsp14 MTase. Protein backbones are shown in the cartoon representation and depicted in yellow (monkeypox virus VP39), magenta (SARS-CoV-2 nsp14). TO507 and the side chains of selected residues are shown in the stick representation with carbon atoms colored according to the protein assignment and other elements colored as in Fig. 1b.

probably important to disrupt the water network, but it is not important to replace it with other hydrogen bonds.

## Comparison to other viral MTases

Finally, we compared the catalytic sites of VP39 with the catalytic sites of 2′-O-ribose MTases from unrelated medically important viruses, namely SARS-CoV-2 and Zika. The resemblance of the SAM binding sites is remarkable in both cases. Sinefungin is virtually in the same conformation in MPXV VP39, SARS-CoV-2 nsp16 and Zika NS5 (Supplementary Fig. 3). The catalytic tetrad (Lys41, Asp138, Lys175, Glu218 for MPXV) is absolutely conserved among these unrelated viruses including the conformation of these residues (Supplementary Fig. 3). Nsp16, the coronaviral 2′-O MTase, does not contain any cavity that could be exploited by the moieties attached at the 7-deaza position. However, nsp14, the coronaviral N7-MTase, contains such a cavity and the cavity is largely hydrophobic[19]. Indeed, some compounds such as TO507 were already identified as nanomolar inhibitors of SARS-CoV-2 nsp14 and docking studies using the previously published structure of nsp14 (pdb entry 7R2V[20]) revealed that the large aromatic substituent at the 7-deaza position is located within this nsp14 cavity (Fig. 5)[19].

## Discussion

Viruses are a global threat to human health as illustrated by the recent COVID-19 pandemic. Even viral diseases with mortality rates ~ 1% can kill millions of people if they spread globally. Monkeypox is already present on every continent, its spread was very likely facilitated by cessation of the vaccination against smallpox in the 1980s since the smallpox vaccine is effective against the monkeypox (~80% efficiency)[21]. With an estimated mortality rate of 3–6%, the MPXV has the potential to cause a devastating pandemic[22]. On the other hand, the effective FDA-approved vaccine JYNNEOS[23] is available, as is the FDA-approved drug - tecovirimat[24]. Encouragingly, MPXV is a dsDNA virus, and the mutation rate of DNA viruses is much lower than that of RNA viruses, such as SARS-CoV-2 or Zika. Nonetheless, it would be wise to develop more drugs to ensure their availability and efficiency.

RNA MTases are present in multiple unrelated viral families, including both RNA and DNA viruses[25]; yet, none of the FDA approved drugs target a viral MTase. Most current antivirals are inhibitors of polymerases, integrases or proteases. In fact, only a handful of inhibitors of viral MTases, mostly targeting flaviviruses[26,27], were known prior to the COVID-19 pandemic. However, recently many inhibitors of SARS-CoV-2 MTases were reported by us and others[19,28–33].

It remains to be established if MPXV MTases are viable antiviral targets; however, the high conservation of VP39 across orthopoxviruses suggests an important role in the viral lifecycle. In addition, the fact that the same compounds can be sub-micromolar inhibitors of both MPXV VP39 and SARS-CoV-2 (Fig. 5) and the similarity of the dsDNA virus VP39 SAM binding site with the 2′-O MTases from two +RNA viral families (flaviviruses and coronaviruses, Supplementary Fig. 3) suggests that the development of a pan-antiviral MTase inhibitor should be feasible, especially since the catalytic tetrad is absolutely conserved. However, to obtain specific antiviral inhibitors that will not interfere with the human RNA 2′-O-MTases, it might be necessary to take advantage of specific features of the viral enzymes, such as a large hydrophobic pocket above the adenine base that is present in coronavirus nsp14[19,34] or the complex water network adjacent to the adenine base in the case of MPXV VP39. Nonetheless, inhibitors of viral MTases are urgently needed to further elucidate the role of these enzymes during viral infection.

## Methods

### Protein expression and purification

The gene for the monkeypox (USA-May22 strain) viral MTase VP39 was codon-optimized for expression in *Escherichia coli* and commercially synthesized (Azenta). Subsequently it was cloned into a pSUMO (with 8xHis-SUMO tag) using restriction cloning (BamHI and NotI sites). The recombinant protein was purified according to our established protocols for viral methyltransferases[35]. Briefly, *E. coli* BL21(DE3) cells were transformed with the VP39 expression plasmid and incubated in LB medium supplemented with $ZnSO_4$ (10 μM) and ampicillin (100 μg/ml) and cultured at 37 °C. When the cells reached optical density 0.6 the production of VP39 was induced by the addition of IPTG (isopropyl-β-D-thiogalactopyranoside) (250 μM) and the temperature was lowered to 20 °C for 15 h. The cells were pelleted by centrifugation and resuspended in lysis buffer (25 mM Tris pH 8, 300 mM NaCl, 30 mM imidazole pH 8, 3 mM β-mercaptoethanol) and lysed by sonication. The clarified lysate was subjected to affinity chromatography on $Ni^{2+}$ agarose resin (Macherey-Nagel) and washed extensively with lysis buffer. The recombinant protein was eluted with lysis buffer supplemented with 300 mM imidazole pH = 8.0. The 8xHis-SUMO tag was cleaved by Ulp1 protease during dialysis against the lysis buffer. Subsequently the Ulp1 and His8x-SUMO tag were removed by a second round of affinity chromatography. Unbound fraction was concentrated and loaded on HiLoad 16/600 Superdex 75 gel filtration column (Cytivia) in size exclusion buffer: 10 mM Tris pH 7.4, 150 mM NaCl, 2 mM β-mercaptoethanol and 5% glycerol. The purified VP39 was concentrated to a set of different concentrations from 7 to 45 mg/ml and directly used for crystallization trials. Coronaviral nsp16 and nsp10 were expressed and purified as before[14] as detailed in Supplementary Information.

## Synthesis of VP39 inhibitors

Compounds TO427, TO494, TO500, TO504, and TO507 were synthesized as described before[19]. All other compounds were synthesized using standard methods of organic synthesis as detailed in the Supplementary Methods. The purity of each compound was verified using an NMR ¹H and ¹³C NMR spectra (Supplementary Figs. 4–8).

## Crystallization and data collection

VP39 protein supplemented with 1 mM sinefungin or 0.6 mM home-made inhibitor (TO427, TO494, TO500, TO507 or TO1119) was plated using the sitting drops method. 200 nl protein solution and 200 nl reservoir solution were mixed in several commercial crystallization screens using a Mosquito robot (SPT Labtech). Only low-quality crystals were generated in the initial experiments, which were then used as crystallization seeds in further experiments. The initial crystals were crushed with a glass rod and diluted 10 000x to produce seeds. The seeding screen was prepared using Dragonfly (SPT Labtech) by gradually decreasing the PEG concentration of the original hit (200 mM lithium citrate, 20% (w/v) PEG 3350). The use of Angstrom Additive Screen TM (Molecular Dimensions) improved the quality of the seed. Finally, thin plate shaped crystals grew within 24 h at 18 °C in 200 mM lithium citrate and 14.5% (w/v) PEG 3350 and were cryoprotected in a 20% (v/v) glycerol-enriched well solution and snap-frozen in liquid nitrogen.

The crystallographic datasets were collected from a single crystal on the BL14.1 beamline at the BESSY II electron storage ring operated by the Helmholtz-Zentrum Berlin[36]. The datasets were collected at the temperature of 100 K using the wavelength of 0.9184 Å. The crystals diffracted in the range of 2–3 Å resolution and belonged to the $P2_12_12_1$ space group. The data was integrated and scaled using XDS[37]. The structure of the MPXV VP39/sinefungin complex was solved by molecular replacement using the structure of the vaccinia virus VP39/SAH complex as a search model (pdb entry 1VP3)[38]. The structures with other inhibitors were solved in a similar fashion except that the structure of MPXV VP39 was used as a search model. The initial models were obtained with Phaser[39] from the Phenix package[40]. The models were further improved using automatic model refinement with Phenix.refine[41] from the Phenix package[40] and manual model building with Coot[42]. Statistics for data collection and processing, structure solution and refinement are summarized in Table 1. Final Ramachandran statistics were as follows: favored in the range 98–100%, allowed in the range 0–2%, outliers 0%. Structural figures were generated with the PyMOL Molecular Graphics System v2.0 (Schrödinger, LLC).

## RNA substrate preparation

For the 2′-O-MTase, an m7GpppG-capped RNA or an m7GpppA-capped RNA were used. These RNAs were 35 nucleotides long and they were prepared by in vitro transcription using a double-stranded DNA template (5′-CAGTAATACGACTCACTATAGGGGAAGCGGGCATGCGGCC AGCCATAGCCGATCA-3′) and the TranscriptAid T7 high-yield transcription kit (Thermo Scientific). The reaction was performed in a 50-µl mixture containing 1× TranscriptAid reaction buffer, 7.5 mM nucleoside triphosphates [NTPs], 1 µg template DNA, and 1× TranscriptAid enzyme mix and 6 mM cap analog m7GpppG (Jena Bioscience) or m7GpppA (prepared chemically according to a published protocol[43]). The reaction was incubated at 37 °C for 8 h. Next, the RNA was purified using RNA Clean & Concentrator-5 from Zymo Research. DNA was removed by DNase I treatment performed directly on the column.

## 2′-O-MTase assay

The reaction was performed in a total volume of 15 µl and all the compounds were diluted in DEPC water to prevent contamination with RNases. The reaction mixture contained 4 µM SAM (S-Adenosyl-L-methionine), 4 µM m7GpppG-capped RNA or m7GpppA-capped RNA in the MTase reaction buffer (5 mM Tris pH 8, 1 mM TCEP, 0.1 mg/ml BSA, 0.005% Triton X-100, 1 mM $MgCl_2$). The reaction was initiated by adding 500 nM VP39, mixed, spun, and incubated at 30 °C for 120 min. The reactions were stopped by adding 30 µl of 7.5% formic acid and analyzed on an Echo mass spectrometry system coupled with a Sciex 6500 triple-quadrupole mass spectrometer operating with an electrospray ionization source. The rate of MTase activity was measured as the amount of the product of the reaction SAH. The spectrometer was run in the multiple-reaction-monitoring (MRM) mode with the interface heated to 350 °C. The declustering potential was 20 V, the entrance potential 10 V, and the collision energy 28 eV. Ten nanoliters was injected in the mobile phase (flow rate of 0.46 ml/min; 70% acetonitrile with 0.1% formic acid). The characteristic product ion of SAH, m/z 385.1 > 134.1, was used for quantification.

For measurements of inhibition of the 2′-O-MTase reaction the protocol was modified for 1536-well plates. The total volume was 4 µl per well and the reaction mixture contained 4 µM SAM and 4 µM m7GpppG-capped RNA in the MTase buffer. RNase inhibitor (Human Placenta, New England Biolabs) was added to the reaction in the concentration of 1 U/µl to prevent potential RNase degradation. The reaction was initiated by adding 200 nM VP39, mixed, spun, and incubated at 30 °C for 45 min. The reactions were stopped by adding 2 µl of 15% formic acid and analyzed as stated above. For $IC_{50}$ measurements, concentrations of tested compounds were from 0.025 nM to 25 µM and measurements were done in duplicates. Dose-response curves were obtained by fitting the experimental peak area of SAH and different inhibitor concentration. The half maximal inhibitory concentration ($IC_{50}$) was calculated by variable slope model using Graph Pad 8.3.4 software using the equation: $Y = Min + (Max-Min)/(1 + 10^{((LogIC_{50}-X)*HillSlope)})$, where Max, Min are the upper and lower asymptotes.

For measurements of inhibition of the SARS-CoV-2 2′-O-MTase nsp16, a similar protocol was used. The 2′-O-MTase reaction mixture contained 500 nM SARS-CoV-2 nsp16, 8 µM SARS-CoV-2 nsp10 (the activating subunit of nsp16), 4 µM SAM, 4 µM m7GpppA-capped RNA and 1 U/µl of RNase inhibitor in the reaction buffer (5 mM Tris pH 8, 1 mM $MgCl_2$, 3 mM DTT).

## Reporting summary

Further information on research design is available in the Nature Portfolio Reporting Summary linked to this article.

## Data availability

The atomic coordinates and structural factors were deposited in the Protein Data Bank (https://www.rcsb.org) under the PDB accession codes 8B07, 8CGB, 8CEQ, 8CER, 8CES, 8CET, 8CEV. Already published PDB entries 1AV6, 1VP3, and 7R2V were used in this study. Source data are provided with this paper.

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

## Acknowledgements

We thank the Helmholtz-Zentrum Berlin für Materialien und Energie for the allocation of synchrotron radiation beamtime. We are grateful to Dr. Gert Weber and Dr. Frank Lennartz for assistance during crystallographic data collections. We are also grateful to Mike Downey for critical reading of this manuscript. This research was funded by the project the National Institute Virology and Bacteriology (Programme EXCELES, Project No. LX22NPO5103) - Funded by the European Union - Next Generation EU awarded to R.N. and E.B. The project was supported by the Academy of Sciences of the Czech Republic as part of the Strategy AV 21 Virology and Antiviral Therapy programme awarded to R.N. and E.B.; RVO: 61388963 is also acknowledged.

## Author contributions

J.S., M.K., T.O., P.S., D.C., K.C., and J.K. performed experiments. J.S. and M.K. analyzed data, J.S., M.K., T.O., P.S., and K.C. prepared the figures. R.N. and E.B. designed experiments and supervised the project. E.B. and J.S. conceived the project and E.B. wrote the manuscript.

## Competing interests

The authors declare no competing interests.

 
