## [Peer Review File · Nature Communications]

REVIEWER COMMENTS

Reviewer #1 (Remarks to the Author):

An interesting and straightforward paper from Boura and colleagues describing the crystal structure of 2'-O-ribose methyltransferase (MTase) VP39 from the dsDNA monkeypox virus. Monkeypox has emerged as a major health concern over the past year and there is a great deal of interest in developing new antivirals against this virus. The paper presents the first structure, to my knowledge, of the 2'-O-ribose MTase from the virus, and for that reason is meritorious of publication in the journal. The structure is well resolved (at 2 Angstrom resolution) and shows interactions that the MTase makes with the general inhibitor sinefungin; generally, very similar to those observed with MTases from ssRNA virus such as Zika and SARS-CoV-2.

Comments:

1. On page 5, I think the authors meant to say "It remains to be established if MPXV MTases are viable antiviral targets...."
2. Figures 3 and 5 could easily be move to supplementary, and figure 4 could easily be a sub-panel of figure 2.
3. The overlay between the active sites in figure 5 is quite confusing. I think it would be better to display each structure separately (in supplementary).

Reviewer #2 (Remarks to the Author):

This manuscript describes a structure of monkeypox 2'-OH-ribose methyltransferase VP39 in complex with a non-specific inhibitor sinefungin. This enzyme is involved in the modification of the first nucleotide of the capped viral mRNA which is important for the evasion of the innate immune response of the infected cell. No clinically useful inhibitors of viral RNA methyltransferases are currently available.

Technically, this is a very solid work. The structure is solved at high resolution of ~2 Å with good statistics. The validation report shows that the structure is of high quality. One issue are the protein residues with high RSRZ scores (poor fit to the electron density). Does the density support the modeling of these residues?

As far as the general impact and importance of this work is concerned, its main strength is that the studied enzyme is from an emerging pathogen, which recently attracted a lot of attention worldwide. However, this work presents only a single structure and general novel insights are rather limited. The structure is very similar to the previously determined structures of viral 2'-OH-ribose methyltransferases. The mode of sinefungin binding is also nearly the same. Moreover, sinefungin is not a useful lead compound because it is a non-specific inhibitor. This limits the usefulness of the presented structural information for the development of a specific inhibitor.

There are different ways to increase the impact and novelty of this work. This could for example include determination of the structures with cap analogs to elucidate the mechanism of the intriguing nucleotide preference (Fig. 1C), determination of the structures with potential specific lead compounds or use of the available crystals for fragment-based screening approaches. I admit that all of these would require significant additional effort which is most likely beyond the time frame of the revision of the current work.

Minor comments.

The first part of the results section does not mention that the crystals were grown in the presence of sinefungin, so the description of the inhibitor binding comes as a surprise.

Please specify the type of omit map shown in Fig 1B.

REVIEWER COMMENTS

Reviewer #1 (Remarks to the Author):

An interesting and straightforward paper from Boura and colleagues describing the crystal structure of 2'-O-ribose methyltransferase (MTase) VP39 from the dsDNA monkeypox virus. Monkeypox has emerged as a major health concern over the past year and there is a great deal of interest in developing new antivirals against this virus. The paper presents the first structure, to my knowledge, of the 2'-O-ribose MTase from the virus, and for that reason is meritorious of publication in the journal. The structure is well resolved (at 2 Angstrom resolution) and shows interactions that the MTase makes with the general inhibitor sinefungin; generally, very similar to those observed with MTases from ssRNA virus such as Zika and SARS-CoV-2.

We appreciate the Expert Reviewer #1 believes that our study is interesting and straightforward.

Comments:

1. On page 5, I think the authors meant to say "It remains to be established if MPXV MTases are viable antiviral targets...."

Indeed, we apologize. The mistake was corrected.

Now we state (page 6, last paragraph):

"It remains to be established if MPXV MTases are viable antiviral targets; however, the high conservation of VP39 across orthopoxviruses suggests an important role in the viral lifecycle."

2. Figures 3 and 5 could easily be move to supplementary, and figure 4 could easily be a sub-panel of figure 2.

The Figures were substantially reorganized due to inclusion of a lot of new data. Figure 3 is now a subpanel in Figure 2 (Figure 2B). Figure 4 is now Figure 3A. And finally Figure 5 was moved to SI.

No we show these figures:

Fig. 2 Sinefungin and SAH recognition by the monkeypox virus VP39 methyltransferase. **a** Sinefungin is bound in the SAM binding site. VP39 is shown in surface representation. **b** Detailed view of sinefungin or **d** SAH in the binding site. Sinefungin, SAH and side chains of selected VP39 amino acid residues are shown in the stick representation with carbon atoms colored according to the protein/ligand assignment and other elements colored as usual. Selected hydrogen bonds involved in the VP39-sinefungin interaction are presented as dashed black lines. **c** Model of RNA recognition by the monkeypox virus VP39 MTase, surface of the VP39 protein is colored according to the electrostatic surface potential. The m7GpppG-capped RNA was modeled by structural alignment using the crystal structure of the vaccinia virus VP39 protein in complex with SAH and m7GpppG-capped RNA (pdb entry 1AV6) as a template. Sinefungin and m7GpppG-capped RNA are shown in the stick representation and colored according to the elements.

Fig. 3 A network of coordinated water molecules in the vicinity of the sinefungin binding groove is disturbed by inhibitor binding. **a** Detailed view of the water molecules coordinated next to the adenine base of sinefungin. Sinefungin and the side chains of selected VP39 amino acid residues are shown in the stick representation with carbon atoms colored according to the protein/ligand assignment and other elements colored as in Fig 1B. Water molecules are shown as red spheres; selected hydrogen bonds are presented as dashed black lines. **b - f** The same view for different inhibitors.

3. The overlay between the active sites in figure 5 is quite confusing. I think it would be better to display each structure separately (in supplementary).

We agree with the Expert Reviewer #1 that the superposition figure might be confusing and we now display each structure separately as suggested. We now only include the original figure in supplementary as SI Figure 3 for an interested reader.

SI Fig. 3 Structural alignment of the methyltransferase active sites of monkeypox virus VP39 with SARS-CoV-2 nsp16 and Zika virus NS5. Crystal structures of SARS-CoV-2 nsp10/nsp16 (a, pdb entry 6yz1) and Zika virus NS5 methyltransferase domain (b, pdb entry 5mrk) were used for the alignment. Protein backbones are shown in the cartoon representation and depicted in yellow (monkeypox virus VP39), magenta (SARS-CoV-2 nsp16), and green (Zika virus NS5). Sinefungin and the side chains of selected residues are shown in the stick representation with carbon atoms colored according to the protein assignment and other elements colored as in Fig 1B.

Reviewer #2 (Remarks to the Author):

This manuscript describes a structure of monkeypox 2'-OH-ribose methyltransferase VP39 in complex with a non-specific inhibitor sinefungin. This enzyme is involved in the modification of the first nucleotide of the capped viral mRNA which is important for the evasion of the innate immune response of the infected cell. No clinically useful inhibitors of viral RNA methyltransferases are currently available.

Technically, this is a very solid work. The structure is solved at high resolution of ~2 Å with good statistics. The validation report shows that the structure is of high quality. One issue are the protein residues with high RSRZ scores (poor fit to the electron density). Does the density support the modeling of these residues?

We appreciate that the Expert Reviewer #2 concludes that our work is solid and that the structure is of high quality.

Below is shown an Fo-Fc omit map contoured at 1.5σ around the top six RSRZ outliers, i.e. all residues with $RSRZ > 5$. We believe that the density supports the modeling of these residues, nevertheless, we admit that more alternative conformations may be possible that cannot be modeled due to the poor electron density. We would prefer to keep these sidechains in the model but we are prepared to delete them if the Expert Reviewer #2 thinks it would be more appropriate. In that case, we would also delete similar residues in our new structures with inhibitors.

Figure for the Expert Reviewer #2: An Fo-Fc omit map contoured at 1.5σ around the top six RSRZ outliers.

As far as the general impact and importance of this work is concerned, its main strength is that the studied enzyme is from an emerging pathogen, which recently attracted a lot of attention worldwide.

We appreciate that the Expert Reviewer #2 identified the main strength of our manuscript.

However, this work presents only a single structure and general novel insights are rather limited. The structure is very similar to the previously determined structures of viral 2'-OH-ribose methyltransferases.

We have added six more crystal structures, new compounds and biochemical data (details below) to strengthen our manuscript.

The mode of sinefungin binding is also nearly the same. Moreover, sinefungin is not a useful lead compound because it is a non-specific inhibitor. This limits the usefulness of the presented structural information for the development of a specific inhibitor.

We agree with the Expert Reviewer #2 that a structure with a specific inhibitor is a better starting point. However, we were previously able to utilize structure of a coronaviral nsp14 MTase in complex with sinefungin/SAH to generate nanomolar inhibitors (Otava et al. *The Structure-Based Design of SARS-CoV-2 nsp14 Methyltransferase Ligands Yields Nanomolar Inhibitors*. ACS Infect Dis. 2021).

We invited our medicinal chemistry collaborator - the Nencka lab - to help us on this project. We have optimized our VP39 MTase assay for high-throughput and we have screened our preexisting mini-library of MTase inhibitors and we also synthesized new compounds. We have identified almost ten compounds that are significantly better inhibitors of the mpox VP39 than sinefungin. (Please, note the IC₅₀ value for sinefungin has changed because IC₅₀ values strongly depend on the conditions of the assay).

There are different ways to increase the impact and novelty of this work. This could for example include determination of the structures with cap analogs to elucidate the mechanism of the intriguing nucleotide preference (Fig. 1C), determination of the structures with potential specific lead compounds or use of the available crystals for fragment-based screening approaches. I admit that all of these would require significant additional effort which is most likely beyond the time frame of the revision of the current work.

We appreciate that the Expert Reviewer #2 believes that these additional experiments would require significant additional effort which is most likely beyond the time frame of the revision. We were able to include these new compounds and data only because of the help of the Nencka lab that specializes in medicinal chemistry.

Specifically, we have included six new crystal structures (and biochemical data as mentioned above); five of them in complex with inhibitors and one in complex with SAH. All the inhibitors utilize the water-filled cavity that we identified thanks to the structure with sinefungin. To our surprise even inhibitors that cannot make hydrogen bonds within the cavity (because the substituent is a large aromatic cycle) are much better than sinefungin and the large aromatic cycles are located within the cavity as illustrated by our crystal structures.

Minor comments.

The first part of the results section does not mention that the crystals were grown in the presence of sinefungin, so the description of the inhibitor binding comes as a surprise.

We apologize for the omission. It was corrected. We now state in the Results section (page 3, the very beginning of Results section):

" Overall structure of the MPXV VP39 MTase - To obtain the structure, we expressed and purified recombinant MPXV VP39 in *E. coli*. The recombinant protein was supplemented with sinefungin and screening of the crystallization conditions was performed."

And we also state in the Materials&Methods section (page 7, Crystallization and data collection paragraph):

"*Crystallization and data collection* - VP39 protein supplemented with 1 mM sinefungin or 0.6 mM home-made inhibitor (TO427, TO494, TO500, TO507 or TO1119) was plated using the sitting drops method."

Please specify the type of omit map shown in Fig 1B.

It is a simple omit map calculated with model phases. Therefore, the word "unbiased" is probably an overstatement and is now omitted, we just wanted to emphasize that Fo-Fc is less biased than 2Fo-Fc.

We now state in the legend of Figure 1B:

" **b** Overall fold of the VP39 protein in complex with sinefungin. The protein backbone is shown in cartoon representation and depicted in yellow, while sinefungin is shown in the stick representation and colored according to its elements: carbon, green; nitrogen, blue; oxygen, red. The Fo-Fc omit map contoured at 3σ is shown around sinefungin."

REVIEWERS' COMMENTS

Reviewer #2 (Remarks to the Author):

The authors have addressed my comments appropriately. The inclusion of the new structural and activity data for submicromolar inhibitors of VP39 definitely strengthens this paper.

Regarding the residues with weak density, it is ultimately up to the authors to decide whether to keep them in the structure.

Below are a few minor editorial comments.

Val116 is not labeled in Fig. 2B, D

In SI Fig. 2 both ligands and maps are green which makes them difficult to see.

Line 129. There is a reference to SI Fig 1D - no such panel is present in the manuscript.

Fig. 5B - Please indicate in the legend that this nsp14 model is a result of docking calculations.

REVIEWERS' COMMENTS

Reviewer #2 (Remarks to the Author):

The authors have addressed my comments appropriately. The inclusion of the new structural and activity data for submicromolar inhibitors of VP39 definitely strengthens this paper.

Regarding the residues with weak density, it is ultimately up to the authors to decide whether to keep them in the structure.

We are grateful to Expert Reviewer #2 for his comments and suggestions.

Below are a few minor editorial comments.

Val116 is not labeled in Fig. 2B, D

Now, Val116 is labeled in both Fig 2B and D.

In SI Fig. 2 both ligands and maps are green which makes them difficult to see.

Now, the maps in SI Fig. 2 are colored in grey.

Line 129. There is a reference to SI Fig 1D - no such panel is present in the manuscript.

Indeed, we apologize for the mistake. The corresponding sentence referring to a non-existent figure and to data not presented in this manuscript was deleted.

Fig. 5B - Please indicate in the legend that this nsp14 model is a result of docking calculations.

Now, the legend of Fig 5B has been changed to: "Docking model of inhibitor TO507 bound to SARS-CoV-2 nsp14 MTase."